# Small-Scale Livestock Production in Nepal Is Directly Associated with Children’s Increased Intakes of Eggs and Dairy, But Not Meat

**DOI:** 10.3390/nu12010252

**Published:** 2020-01-18

**Authors:** Elena T. Broaddus-Shea, Swetha Manohar, Andrew L. Thorne-Lyman, Shiva Bhandari, Bareng A. S. Nonyane, Peter J. Winch, Keith P. West

**Affiliations:** 1Center for Human Nutrition and Department of International Health, Johns Hopkins Bloomberg School of Public Health, 615 N. Wolfe St., Baltimore, MD 21205, USA; smanoha2@jhu.edu (S.M.); athorne1@jhu.edu (A.L.T.-L.); bnonyan1@jhu.edu (B.A.S.N.); pwinch1@jhu.edu (P.J.W.);; 2Department of Family Medicine, University of Colorado Anschutz Medical Campus, 13001 E. 17th Place, Aurora, CO 80045, USA; 3Department of Health Promotion, Education and Behavior, Arnold School of Public Health, University of South Carolina, 915 Greene St., Columbia, SC 29208, USA; bhandarishv@gmail.com

**Keywords:** animal source foods, livestock husbandry, agriculture-nutrition pathways, child nutrition, dietary diversity, Nepal

## Abstract

Animal source foods (ASF) provide nutrients essential to child growth and development yet remain infrequently consumed in rural Nepal. Agriculture and nutrition programs aim to increase ASF intake among children through small-scale animal husbandry projects. The relationship between livestock ownership and children’s consumption of ASF, however, is not well established. This study examined associations between livestock ownership and the frequency with which Nepali children consume eggs, dairy, and meat. We analyzed longitudinal 7-day food frequency data from sentinel surveillance sites of the Policy and Science of Health, Agriculture and Nutrition (PoSHAN) study. Data consisted of surveys from 485 Nepali farming households conducted twice per year for two years (a total of 1449 surveys). We used negative binomial regression analysis to examine the association between the number of cattle, poultry, and meat animals (small livestock) owned and children’s weekly dairy, egg, and meat intakes, respectively, adjusting for household expenditure on each food type, mother’s education level, caste/ethnicity, agroecological region, season, and child age and sex. We calculated predicted marginal values based on model estimates. Children consumed dairy 1.4 (95% CI 1.1–2.0), 2.3 (1.7–3.0) and 3.0 (2.1–4.2) more times per week in households owning 1, 2–4 and >4 cattle, respectively, compared to children in households without cattle. Children consumed eggs 2.8 (2.1–3.7) more times per week in households owning 1 or 2 chickens compared to children in households without chickens. Child intake of meat was higher only in households owning more than seven meat animals. Children’s intakes of dairy, eggs, and meat rose with household expenditure on these foods. Small-scale animal production may be an effective strategy for increasing children’s consumption of eggs and dairy, but not meat. Increasing household ability to access ASF via purchasing appears to be an important approach for raising children’s intakes of all three food types.

## 1. Introduction

Animal source foods (ASF) provide essential nutrients for healthy cognitive and physical development of young children [1,2,3,4]. However, in subsistence agriculture settings like rural Nepal, such foods are often scarce and infrequently consumed [5,6,7]. To improve the availability and consumption of ASF, interventions promoting small-scale livestock production are increasingly being implemented in Nepal and other low-income rural settings [8,9,10]. Examples include Helen Keller International’s poultry production program model and Heifer International’s livestock donation and management training program model [11,12].

Evaluations of the impact of livestock husbandry interventions on child diet and nutritional status have shown mixed results [11,12,13,14,15,16,17]. Similarly, observational studies of the relationship between livestock ownership and child dietary intake suggest that the relationship is complex and highly dependent on contextual factors that determine how livestock and home-produced ASF are utilized [18,19,20,21]. Several recent reviews of the agriculture-nutrition literature conclude that a more detailed understanding of how livestock ownership is related to children’s ASF consumption is needed in order to determine if and how small-scale livestock production programs can be leveraged to improve child nutrition [15,22,23]. Conceptual frameworks of agriculture to nutrition linkages have outlined both direct pathways from production to consumption and indirect pathways that involve the sale of produced foods and purchase of other foods using the income generated [24,25]. However, the relative importance of these pathways for improving child diets is not well understood. Specifically, for ASF, little is known about what scale of animal production is needed to diversify children’s diets, whether this depends on the types of animal source foods produced, and the relative importance of access to home-produced ASF via livestock ownership vs. market access [22].

This study aims to understand (1) whether household livestock ownership is directly related to child consumption of eggs, dairy products, and meat, controlling for expenditures on those foods, and (2), how the amounts of poultry, cattle, and small livestock owned by households are associated with child consumption of eggs, dairy products, and meat.

## 2. Materials and Methods

Data were collected in three rural Nepali communities (one each in the mountains, hills, and plains) as part of the larger Policy and Science of Health, Agriculture, and Nutrition (PoSHAN) study, described in detail elsewhere [26,27,28,29,30,31]. Within each village development committee (VDC), three wards (the smallest administrative unit in Nepal) were randomly selected and, within those wards, all households that contained a child under the age of six years, or a pregnant or newly married woman, were invited to participate in the study. Households were surveyed thrice yearly, to account for seasons, over a period of two years. Two survey rounds conducted between September and February each year comprised unique seasonal assessments, while one survey round conducted between May and July each year was undertaken as part of nationally scheduled PoSHAN surveys during those months. Data collected included agricultural practices and livestock ownership, household economic and expenditure information, and women’s and children’s dietary intake measured using a 7-day food frequency questionnaire (FFQ) of 51 commonly consumed foods. The FFQ was adopted from a previous tool used specifically within the Nepali context to reflect usual intake, which was further pre-tested and adapted for use in this study [32,33]. Informed consent was obtained from all participants prior to administering each survey. Ethical approval for the PoSHAN study was granted by the Institutional Review Board at The Johns Hopkins Bloomberg School of Public Health and from the Nepal Health Research Council.

The analysis presented here used data from four timepoints: the 2013 and 2014 rainy seasons (May–July) and the 2014 and 2015 dry seasons (January–February). Criteria for inclusion in this analysis included being a farming household (defined as a household that reported growing at least one field crop in the previous year) and having a child between 6 and 72 months in the household. One child was randomly selected as the index child in cases where the household contained two or more 6–72-month-old children. This yielded an analysis sample of 485 households surveyed up to 4 times each for a total of 1449 observations.

Three different regression analyses were conducted to assess: (1) the relationship between poultry ownership and children’s egg consumption frequency; (2) the relationship between meat animal ownership and children’s meat consumption frequency; and (3) the relationship between cattle ownership and children’s dairy consumption frequency. Outcome variables were the number of times that a child consumed foods falling into each of these three food types in the seven days prior to the survey. These variables were generated using the FFQ described above; for dairy and meat the frequencies of consumption for all relevant foods on the FFQ were summed to generate the outcome variables. The main predictor variables of interest were the number of poultry, meat animals, and cattle that a household owned categorized as none, low, medium, or high for each livestock type (using cut-points based on the inter-quartile range for households with any livestock ownership). Our definition of meat animals included animals commonly raised specifically for consumption in Nepal (i.e., not cows, buffalo, yak, or horses): poultry, goats/sheep, guinea fowl/pigeons, and pigs. Our definition of cattle included cows, buffalo, and yak—animals which are commonly used for milk production in Nepal, in addition to serving a variety of other agricultural purposes. In order to adjust for household expenditure on eggs, meat, and dairy, 30-day household expenditures for foods within each category were summed and then categorized as none, low, medium or high for each food type (again, using cut-points based on the inter-quartile range for households with any expenditure). Each analysis also adjusted for child-specific, demographic, and geographic factors known to influence household food access and/or child diet, including child age, child sex, mother’s education level, caste/ethnicity, region (mountains, hills, or plains (*Tarai*)), and season. Missing values were negligible (less than 0.5% of observations) and a complete-case analysis was conducted.

Summary statistics of the participant characteristics and of the outcome variables were generated and negative binomial generalized-estimating equation (GEE) models were used to generate unadjusted and adjusted population-averaged incident rate ratios (IRR), accounting for the correlation among repeated measures on the same household. All analyses were conducted in STATA 14 [34]. Analysis results were interpreted using a significance cut-off of *p* < 0.05. To ease interpretation, final model estimates were used to generate and plot predicted marginal values of each outcome at varying levels of livestock ownership and ASF expenditure.

## 3. Results

Children’s consumption of ASF was very low, with most not consuming any eggs or dairy within the seven days prior to each survey date, and median meat consumption per week was 1.0 (Table 1). For the majority of observations, households owned cattle (71%) and meat animals (67%), and for 36% of observations households owned poultry. Expenditure on meat was highest compared to the other food groups, with 83% of observations indicating that households purchased meat in the 30 days prior to the survey; these households spent a median amount of 1000 NPR on meat (equivalent to approximately 10 US dollars). Far fewer households purchased eggs or milk (35% of observations and 30% of observations, respectively), with median monthly expenditure of 400 NPR for milk and 180 NPR for eggs.

Adjusted GEE models (Table 2) indicated that low, medium, or high poultry ownership were associated with higher children’s egg intake compared to no ownership. Similarly, low, medium, and high levels of cattle ownership were associated with higher levels of dairy intake compared to no ownership. Meat intake was associated with only a high level of livestock ownership compared to no ownership. As the estimated marginal values plotted in Figure 1 show, the relationship between increasing levels of livestock ownership and children’s intakes differed across the three food types. Ownership of just one or two poultry was associated with children consuming eggs a predicted average of 2.2 (95% CI 1.7–2.8) times per week, while children in households without poultry were predicted to consume eggs less than once per week. However, at higher levels of poultry ownership, findings indicated a leveling off—i.e., children in households owning three or more poultry were not predicted to consume significantly more eggs compared to those in households owning just one or two poultry. For dairy, findings suggested a monotonic relationship between level of cattle ownership and children’s dairy intakes. Children in households owning zero, one, two to four, or more than four cattle were predicted to consume dairy an average of 3.1 (95% CI 2.4–3.9), 4.5 (95% CI 3.6–5.4), 7.1 (95% CI 6.1–8.1), and 9.3 (95% CI 7.0–11.5) times per week, respectively. For meat, only the highest level of livestock ownership (more than seven) was associated with an increase in children’s meat intake, and the estimated effect was small—children in households owning more than seven meat animals were predicted to consume meat an average of 1.6 (95% CI 1.3–1.8) times per week, while children in households without any meat animals were predicted to consume meat an average of 1.2 (95% CI 1.0–1.4) times per week.

Household expenditures on each type of ASF were strongly associated with intake of those foods. As shown in Table 2 and Figure 1, the multivariable-adjusted analyses and marginal plots indicate that greater levels of expenditure were associated with higher consumption frequencies for all food types. The only exception was low monthly expenditures on dairy (NPR 1–180 = USD 0.01–1.8), which were not associated with an increase in children’s intake compared to children in households with no expenditure on dairy in the past month. However, both medium and high levels of expenditure were associated with significantly higher intake levels.

## 4. Discussion

Our study suggests that in rural Nepal, poultry and cattle ownership, even at low levels, is associated with higher intakes of eggs and dairy, respectively, among young children. These findings provide further empirical support for programs that promote small-scale animal production as a potentially effective strategy for increasing children’s consumption of eggs and dairy [12,13,14,17]. They are broadly consistent with recent observational studies of the positive relationship between cattle ownership and children’s dairy intake conducted in other rural low-income countries [19,20,35], and provide a valuable counterpoint to several studies indicating little relationship between poultry ownership and egg consumption in such settings [16,18,20,21]. These differences in findings point towards the complexity of the relationship between poultry ownership, egg production, and children’s egg consumption and the important role of myriad context-specific factors in determining how households utilize poultry and eggs. Indeed, our findings also indicate a complex relationship. Further research is needed to understand the leveling-off that we observed between poultry ownership and children’s egg intakes—i.e., that higher levels of poultry ownership (three or more birds) did not seem to increase egg intake more than a low level of ownership (one to two birds). It may be that households owning more than two birds tend to sell the additional eggs produced or may be raising poultry for meat purposes rather than egg production, whereas households with just one or two birds raise them primarily for in-home egg consumption.

In contrast to the relationship we found between even low levels of livestock ownership and increased egg and dairy intake, increases in meat intake only seem to occur when a household raises a large number (seven or more animals) of livestock commonly used for meat purposes (i.e., poultry, goats/sheep, guinea fowl/pigeons, or pigs). Even then, the predicted effect is quite small—equivalent to less than one additional time consuming meat per week. This is unsurprising given that only those families owning a large number of animals (likely substantially more than seven) will be able to consume meat from their own livestock regularly; instead most families purchase the meat they consume. Indeed, the survey results showed that the majority of households had purchased meat in the past month, while only one-third purchased eggs and dairy. However, raising meat animals may indirectly increase children’s meat consumption if households sell the animals that they raise and then use the money earned to purchase meat [14]. A recent review of studies examining the relationship between production diversity (including livestock ownership) and dietary diversity among smallholder farming households suggests that the relationship depends largely on the household’s degree of market orientation, and that increasing affordable market access to a diverse range of foods may be more important for improving diets than diversifying home-produced foods [22]. For the households in this study, income generated by raising and then periodically selling meat animals may enable higher levels of expenditure on a variety of foods and other important goods and services. Further analyses are needed to understand how these households utilize meat animals and any income earned through their sale. However, to increase meat consumption, small-scale animal production programs would likely need to have a substantial income generation emphasis. The strong association between increasing household expenditure and children’s ASF intakes suggests that enabling households to increase their monthly ASF expenditure or subsidizing households to access such foods may be important strategies for increasing children’s consumption of eggs and dairy, and would be the primary pathway for increasing children’s consumption of meat.

Our findings also reflect differences in children’s ASF intake by caste/ethnicity in the cases of eggs and dairy, and also, by region and season in the case of dairy. These findings are expected and concur with the previous literature on dietary variation in Nepal [36,37,38].

While this study focused on ASF intake as the outcome, further research is needed to determine the levels of ASF consumption by children that may have an impact on different aspects of their nutritional status. This threshold remains uncertain and is likely to vary by indicator and by context. Trials testing the effects of daily egg provision to young children have found mixed results, with one study in Ecuador initially finding a marked impact on linear growth but with little evidence of lasting benefit at two years post-intervention, and another from Malawi finding no effect on linear growth [39,40,41]. Study authors note that this may point towards a need for longer intervention periods in the case of the Ecuador trial, and that in Malawi, the intervention’s potential impact may have been limited by already-ASF-rich diets and low stunting prevalence at baseline. In Nepal, the likely potential benefit of increased ASF consumption is substantial (given high stunting prevalence and low ASF intake). However, questions remain as to how much additional consumption of eggs or milk by children is needed to improve child growth, development, or micronutrient status, and what degree of animal ownership by households or by communities is needed to assure that level of consumption.

Limitations of this analysis include potential error in food frequency data due to inaccurate recall, and lack of data on ASF obtained via sharing, bartering, and hunting. However, a similar 7-day FFQ approach has been used in the *Tarai* of Nepal and results of maternal dietary intake were found to reflect seasonal patterns and correlate with socioeconomic status [32,33]. Another limitation is that the recall period for dietary intake data was 7 days, while that for household food expenditures was 30 days, providing an imperfect alignment and potential for error in adjustment for expenditures. Additionally, the analysis approach taken (specifically, adjusting for ASF expenditure levels) means that we were unable to investigate the income generation pathways noted above that undoubtedly are also important routes through which livestock ownership influences children’s ASF intakes. However, by adjusting for ASF expenditure, we were able to examine the direct association between livestock ownership and children’s ASF consumption. These findings make a valuable contribution to the existing literature on livestock production and diet in low-income rural settings. They provide empirical support for small-scale livestock husbandry programs as a strategy to improve children’s consumption of eggs and dairy products in Nepal.

## Figures and Tables

**Figure 1 nutrients-12-00252-f001:**
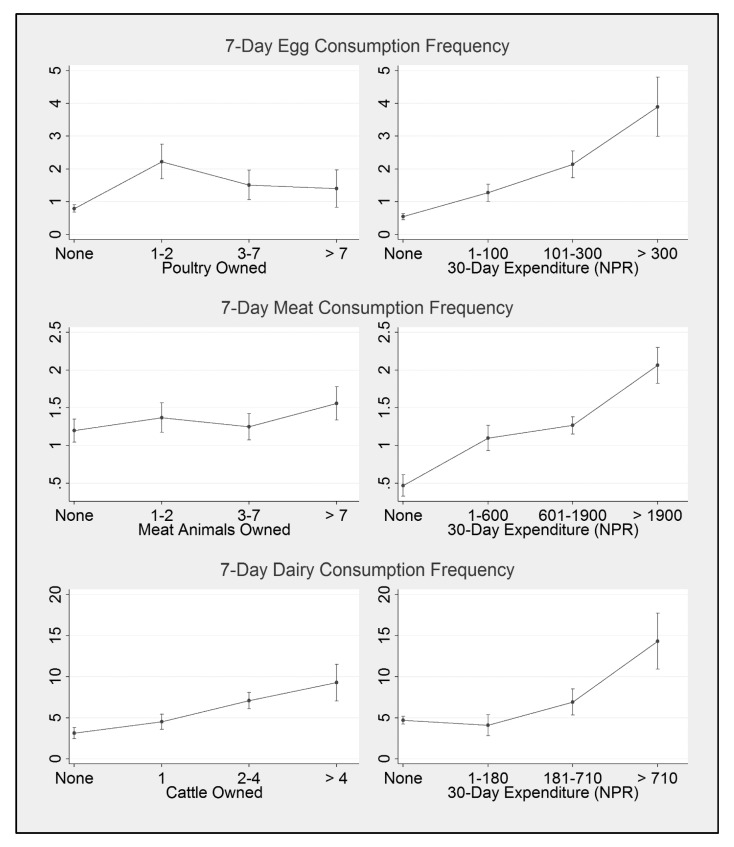
Predicted average weekly consumption frequency (with 95% confidence intervals) of eggs, meat, and dairy at different levels of animal production and expenditure on animal-source foods, adjusting for all other covariates (mother’s education, caste/ethnicity, region, season, child sex, child age). Predictions were generated using negative binomial model estimates presented in Table 2.

**Table 1 nutrients-12-00252-t001:** Characteristics of Nepali subsistence farming households observed at up to four different time points.

Time-Invariant Characteristics	
Total Households (*N*)	485
Region (N (%))	
Mountains	132 (27.2%)
Hills	161 (33.2%)
Tarai	192 (39.6%)
Caste/Ethnicity (N (%))	
Dalit	119 (24.5%)
Disadvantaged Janajatis	35 (7.2%)
Disadvantaged Tarai Castes	93 (19.2%)
Religious Minorities	73 (15.1%)
Advantaged Janajatis	22 (4.5%)
Upper Caste	143 (29.5%)
Age in Months of Index Child at Baseline (mean (SD))	31.3 (17.1)
Sex of Index Child (N (%) female)	198 (40.8%)
Mother’s Education Level (N (%))	
None	240 (49.5%)
Primary Level (at least partially completed)	84 (17.3%)
Secondary Level or Higher (at least partially completed)	161 (33.2%)
Time Variant Characteristics	
Total Observations (n)	1449
Index Child’s Weekly ASF Consumption Frequency (median (IQR))	
Dairy	0 (0–7)
Eggs	0 (0–2)
Meat	1 (0–2)
Household Animal Ownership Among Producing Households (median (IQR))	
Number Cattle Among Households with Any Cattle Production	2 (1–4)
Number Poultry Among Households with Any Poultry Production	3 (2–7)
Number Meat Animals Among Households with Any Meat Animal Production	4 (2–7)
Levels of Household Animal Production (n (%)) ^1^	
Cattle Ownership	
None	531 (36.7%)
Low (1 cow or buffalo)	188 (13.0%)
Medium (2–4 cattle)	552 (38.1%)
High (>4 cattle)	178 (12.3%)
Poultry Ownership	
None	932 (64.3%)
Low (1–2 birds)	220 (15.2%)
Medium (3–7 birds)	173 (11.9%)
High (>7 birds)	124 (8.6%)
Meat Animal Ownership	
None	472 (32.6%)
Low (1–2 animals)	322 (21.5%)
Medium (3–7 animals)	435 (29.3%)
High (>7 animals)	241 (16.6%)
Household ASF Expenditure in Past 30 Days in Nepali Rupees (NPR) ^2^ Among Purchasing Households (median (IQR))	
Expenditure on Dairy Among Households Purchasing Dairy	400 (180–710)
Expenditure on Eggs Among Households Purchasing Eggs	180 (100–300)
Expenditure on Meat Among Households Purchasing Meat	1000 (600–1900)
Levels of ASF Expenditure in Past 30 Days (n (%)) ^3^	
Dairy Expenditure	
None	1009 (69.6%)
Low (1–180 NPR)	111 (7.7%)
Medium (181–710 NPR)	219 (15.1%)
High (>710 NPR)	110 (7.6%)
Egg Expenditure	
None	948 (65.4%)
Low (1–100 NPR)	178 (12.3%)
Medium (101–300 NPR)	208 (14.4%)
High (>300 NPR)	115 (7.9%)
Meat Expenditure	
None	240 (16.6%)
Low (1–600 NPR)	335 (23.1%)
Medium (601–1900 NPR)	577 (39.8%)
High (>1900 NPR)	297 (20.5%)

^1^ Cut-offs for categories of animal production were based on the IQR for producer households. ^2^ During the data collection period, the NPR/USD exchange rate was approximately 100 NPR = 1 USD. ^3^ Cut-offs for expenditure categories were based on the IQR for purchaser households.

**Table 2 nutrients-12-00252-t002:** Unadjusted and adjusted negative binomial generalized estimating equation regression results for analyses of the relationship between household livestock ownership and children’s animal source food consumption in 485 Nepali farming households observed at up to 4 timepoints (*n* = 1449 observations).

7-Day Consumption Frequency of Eggs			7-Day Consumption Frequency of Meat			7-Day Consumption Frequency of Dairy		
	Unadjusted	Adjusted ^1^		Unadjusted	Adjusted ^1^		Unadjusted	Adjusted ^1^
	IRR (95% CI) ^2^	IRR (95% CI)		IRR (95% CI)	IRR (95% CI)		IRR (95% CI)	IRR (95% CI)
Poultry Ownership			Meat Animal Ownership			Cattle Ownership		
None	1.0	1.0	None	1.0	1.0	None	1.0	1.0
Low (1–2 birds)	1.6 (1.3–2.1) ***	2.8 (2.1–3.7) ***	Low (1–2 animals)	1.1 (0.9–1.4)	1.1 (0.9–1.4)	Low (1 cow or buffalo)	1.6 (1.2–2.1) ***	1.4 (1.1–2.0) *
Medium (3–7 birds)	1.3 (0.9–1.7)	1.9 (1.4–2.7) ***	Medium (3–7 animals)	1.1 (0.9–1.3)	1.0 (0.9–1.3)	Medium (2–4 cattle)	2.2 (1.8–2.8) ***	2.3 (1.7–3.0) ***
High (>7 birds)	1.3 (0.9–1.9)	1.8 (1.1–2.8) **	High (>7 animals)	1.5 (1.2–1.8) *	1.3 (1.1–1.6) **	High (>4 cattle)	2.7 (2.1–3.5) ***	3.0 (2.1–4.2) ***
Expenditure			Expenditure			Expenditure		
None	1.0	1.0	None	1.0	1.0	None	1.0	1.0
Low (1–100 NPR)	1.8 (1.4–2.4) ***	2.3 (1.8–3.1) ***	Low (1–600 NPR)	2.5 (1.8–3.5) ***	2.3 (1.7–3.2) ***	Low (1–180 NPR)	0.5 (0.4–0.7) ***	0.9 (0.6–1.2)
Medium (101–300 NPR)	3.0 (2.3–3.8) ***	3.9 (3.0–5.1) ***	Medium (601–1900 NPR)	3.3 (2.4–4.4) ***	2.7 (2.0–3.7) ***	Medium (181–710 NPR)	1.0 (0.8–1.2)	1.5 (1.1–1.9) **
High (over 300 NPR)	5.4 (4.0–7.1) ***	7.1 (5.3–9.7) ***	High (over 1900 NPR)	5.5 (4.0–7.5) ***	4.4 (3.2–6.1) ***	High (over 710 NPR)	1.7 (1.4–2.1) ***	3.0 (2.3–4.0) ***
Mother’s Education			Mother’s Education			Mother’s Education		
None	1.0	1.0	None	1.0	1.0	None	1.0	1.0
Primary Level	0.8 (0.6–1.2)	1.0 (0.7–1.5)	Primary Level	1.3 (1.0–1.6) *	1.1 (0.9–1.3)	Primary Level	1.7 (1.2–2.3)**	1.2 (0.8–1.7)
Secondary Level or Higher	1.3 (1.0–1.7)	1.2 (0.9–1.6)	Secondary Level or Higher	1.5 (1.2–1.8) ***	1.2 (1.0–1.5) *	Secondary Level or Higher	2.7 (2.2–3.3) ***	1.3 (1.0–1.9)
Caste/Ethnicity			Caste/Ethnicity			Caste/Ethnicity		
Dalit	1.0	1.0	Dalit	1.0	1.0	Dalit	1.0	1.0
Disadvantaged Janajatis	1.3 (0.8–2.4)	1.0 (0.7–1.5)	Disadvantaged Janajatis	0.9 (0.7–1.1)	0.7 (0.6–0.9) *	Disadvantaged Janajatis	2.5 (1.8–3.5) ***	1.6 (1.1–2.2) *
Disadvantaged *Tarai* Castes	0.7 (0.5–1.1)	0.7 (0.4–1.2)	Disadvantaged *Tarai* Castes	0.3 (0.2–0.4) ***	0.4 (0.3–0.6) ***	Disadvantaged *Tarai* Castes	1.1 (0.8–1.5)	1.1 (0.6–1.7)
Religious Minorities	1.5 (1.1–2.2) *	1.0 (0.6–1.7)	Religious Minorities	0.5 (0.4–0.7) ***	0.7 (0.5–0.9) *	Religious Minorities	0.5 (0.3–0.8) **	0.5 (0.3–0.9) *
Advantaged Janajatis	1.1 (0.7–1.8)	0.9 (0.5–1.5)	Advantaged Janajatis	0.9 (0.7–1.2)	0.8 (0.6–1.0)	Advantaged Janajatis	2.1 (1.4–3.2) ***	1.2 (0.8–1.7)
Upper Caste	1.7 (1.3–2.4) **	1.3 (1.0–1.8)	Upper Caste	0.8 (0.6–0.9) **	0.7 (0.6–0.8) ***	Upper Caste	2.3 (1.7–3.0) ***	1.8 (1.3–2.6) **
Region			Region			Region		
Mountains	1.0	1.0	Mountains	1.0	1.0	Mountains	1.0	1.0
Hills	0.7 (0.6–1.0)	1.0 (0.7–1.4)	Hills	1.2 (1.0–1.5)	1.1 (0.9–1.3)	Hills	3.1 (2.3–4.0) ***	3.2 (2.4–4.4) ***
Plains	0.8 (0.6–1.0)	1.3 (0.8–2.1)	Plains	0.5 (0.4–0.7) ***	1.0 (0.8–1.3)	Plains	1.1 (0.8–1.5)	2.3 (1.4–3.9)**
Season			Season			Season		
Rainy Season	1.0	1.0	Rainy Season	1.0	1.0	Rainy Season	1.0	1.0
Dry Season	1.0 (0.8–1.1)	0.9 (0.8–1.1)	Dry Season	1.0 (0.9–1.1)	1.0 (0.9–1.1)	Dry Season	1.0 (0.9–1.1)	0.8 (0.7–0.9) **

* = *p* < 0.05; ** = *p* < 0.01; *** = *p* < 0.001. ^1^ Also adjusted for child sex and age. ^2^ IRR = Incident Rate Ratio; CI = Confidence Interval.

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
