# Peer review of "Small-Scale Livestock Production in Nepal Is Directly Associated with Children’s Increased Intakes of Eggs and Dairy, But Not Meat"

_nutrients, 2020, doi:10.3390/nu12010252_

Round 1

Reviewer 1 Report

This paper is succinct and well written, with a clear analytic goal that it accomplishes. Although there are already a number of papers examining the relationship between animal ownership and consumption of ASF and/or child growth, this paper adds in two ways: controlling for expenditures and examining the dose-response relationship between animal ownership and consumption frequency. The results are interesting and well presented.

I recommend the paper being published after a few minor changes are made:

Suggest re-framing abstract line 34-35: it is unlikely to be expenditures per se (which could increase, e.g., through an increase in price) but rather access to ASF from off-farm sources (influenced by physical access, affordability (price, income), etc.) There have been a number of papers examining how ASF is associated with child diets and/or nutritional status; the authors may want to review these to ensure they are situating their paper well within the existing literature. Some are already mentioned in the paper, but others are listed below. One take-away message from that prior work (and that on crop diversity and its relationship with dietary diversity, e.g., Sibhatu and Qaim (2018)) has been that the relationship between animal ownership and ASF tends to be mediated by the market: once one is more actively engaged in buying and selling of crops/livestock, the role of on-farm production tends to diminish. This is something for the authors to potentially consider in the discussion, particularly in terms of what the influence of expenditures is capturing. Large livestock often play a more important role in households as assets and income streams, as opposed to sources of food; if this is the case in Nepal, the result on a limited relationship between livestock and meat consumption is unsurprising. If livestock are used as a sign of wealth in Nepal (as in many African countries), the authors might also consider the fact that a large number of large livestock could indicate a wealthy household, which may be more likely to eat meat even if not directly sourced from their own livestock. Did the model adjust for clustering at the ward level? Suggest authors briefly comment on whether the effects seen for caste/ethnicity and region are logical and as expected, given the Nepali context The low percentage of female index children (40%) is surprising; is there a particular reason for this not being close to 50%, or just a statistical anomaly? Does the data allow for looking at livestock ownership by gender, or is all livestock ownership measured at the household level (as opposed to the individual level)? If at the individual level, it might be interesting to examine whether effects are different between men and women owners, given the large literature on gender/women’s empowerment in nutrition-sensitive agriculture as well as on women’s role in livestock production and how ownership relates to control over production (e.g., Njuki et al. 2013). However, I certainly don’t see this extension as necessary for the article to be published—just an avenue to explore if of interest to the authors. Line 171 – might note that one suggested reason for why the trial in Malawi did not result in improved growth was that the background diet was already rich in ASF whereas than in Ecuador was not; which of these settings is the closer analog to Nepal? Please do a proofreading to resolve some minor grammatical / linguistic issues (e.g., line 6 “and Jr.”; comma missing line 72).

Papers related to ASF production and child nutrition:

Bruyn, Julia de, Peter C. Thomson, Ian Darnton-Hill, Brigitte Bagnol, Wende Maulaga, and Robyn G. Alders. 2018. “Does Village Chicken-Keeping Contribute to Young Children’s Diets and Growth? A Longitudinal Observational Study in Rural Tanzania.” Nutrients 10 (11). https://doi.org/10.3390/nu10111799.

Choudhury, Samira, and Derek D. Headey. 2018. “Household Dairy Production and Child Growth: Evidence from Bangladesh.” Economics and Human Biology 30 (September): 150–61. https://doi.org/10.1016/j.ehb.2018.07.001.

Dumas, Sarah E., Lea Kassa, Sera L. Young, and Alexander J. Travis. 2018. “Examining the Association between Livestock Ownership Typologies and Child Nutrition in the Luangwa Valley, Zambia.” PloS One 13 (2): e0191339. https://doi.org/10.1371/journal.pone.0191339.

Dumas, Sarah E., Dale Lewis, and Alexander J. Travis. 2018. “Small-Scale Egg Production Centres Increase Children’s Egg Consumption in Rural Zambia.” Maternal & Child Nutrition 14 Suppl 3 (October): e12662. https://doi.org/10.1111/mcn.12662.

Hanselman, Bailey, Ramya Ambikapathi, Estomih Mduma, Erling Svensen, Laura E. Caulfield, and Crystal L. Patil. 2018. “Associations of Land, Cattle and Food Security with Infant Feeding Practices among a Rural Population Living in Manyara, Tanzania.” BMC Public Health 18 (1): 159. https://doi.org/10.1186/s12889-018-5074-9.

Jones, Andrew D., Esi K. Colecraft, Raphael B. Awuah, Sandra Boatemaa, Nathalie J. Lambrecht, Leonard Kofi Adjorlolo, and Mark L. Wilson. 2018. “Livestock Ownership Is Associated with Higher Odds of Anaemia among Preschool-Aged Children, but Not Women of Reproductive Age in Ghana.” Maternal & Child Nutrition 14 (3): e12604. https://doi.org/10.1111/mcn.12604.

Kabunga, Nassul S., Shibani Ghosh, and Patrick Webb. 2017. “Does Ownership of Improved Dairy Cow Breeds Improve Child Nutrition? A Pathway Analysis for Uganda.” PloS One 12 (11): e0187816. https://doi.org/10.1371/journal.pone.0187816.

Kim, Sunny S., Phuong Hong Nguyen, Lan Mai Tran, Yewelsew Abebe, Yonas Asrat, Manisha Tharaney, and Purnima Menon. 2019. “Maternal Behavioural Determinants and Livestock Ownership Are Associated with Animal Source Food Consumption among Young Children during Fasting in Rural Ethiopia.” Maternal & Child Nutrition 15 (2): e12695. https://doi.org/10.1111/mcn.12695.

Lambrecht, Nathalie J., Mark L. Wilson, and Andrew D. Jones. 2019. “Assessing the Impact of Animal Husbandry and Capture on Anemia among Women and Children in Low- and Middle-Income Countries: A Systematic Review.” Advances in Nutrition (Bethesda, Md.) 10 (2): 331–44. https://doi.org/10.1093/advances/nmy080.

Marquis, Grace S., Esi K. Colecraft, Roland Kanlisi, Bridget A. Aidam, Afua Atuobi-Yeboah, Comfort Pinto, and Richmond Aryeetey. 2018. “An Agriculture-Nutrition Intervention Improved Children’s Diet and Growth in a Randomized Trial in Ghana.” Maternal & Child Nutrition 14 Suppl 3 (October): e12677. https://doi.org/10.1111/mcn.12677.

Other sources mentioned:

Njuki, Jemimah, Elizabeth Waithanji, Joyce Lyimo-Macha, Juliet Kariuki, and Samuel Mburu, eds. 2013. Women, Livestock Ownership and Markets: Bridging the Gender Gap in Eastern and Southern Africa. 1 edition. Abingdon, Oxon ; New York : Ottawa: Routledge.

Sibhatu, Kibrom T., and Matin Qaim. 2018. “Review: Meta-Analysis of the Association between Production Diversity, Diets, and Nutrition in Smallholder Farm Households.” Food Policy 77 (May): 1–18. https://doi.org/10.1016/j.foodpol.2018.04.013.

Reviewer 2 Report

Overall, this is a well-written manuscript with minor errors noted below that need to be corrected.  The reference list is a bit minimal and could be expanded with more recent papers.

Specific comments:

Did the questionnaire ask about gender or sex? Please use the correct term and make adjustment throughout the manuscript. line 56. What is meant by “relative scale”? Missing “and” before (2). line 60. Missing “each” in “one each in the …” Please always use units – line 64. “under the age of six” – six what? months? years? I am a little confused on the frequency of the surveys (lines 65-68). You say “thrice” yet you then mention two time periods. lines 83-84. Cattle does not equate with milking cows. Was a distinction made between milking and non-milking animals? Remove the excess horizontal lines from your tables. Table 1:

- Does “primary level” mean completed or at least some? Same for secondary level.

-Does 1 mean consumed some type of meat one time in last 7 days? What if the child had consumed two types of meat just once – would that be a “2”?

- You need to define the variables in the footnote so it is clear. Define exchange rate.

line 117. You can just say “higher” – drop the “statistically significantly” (it is understood). If it were not significant, then you could not say “higher”.  Include in the methods section what you consider significant. line 119-120 – the sentence reads poorly. line 120 should read “compared” with a “d”. Tables and figures need to be able to stand by themselves – indicate in footnote or figure legend the sample, all abbreviations, analysis used, and covariates included. line 142. Should read “…expenditures… were…”. figure 1 – y axes need a unit. line 154-155. This does not really reflect what is going on with poultry-eggs. Line 157-158. This suggests that families with 7 or more are eating their own animals – I doubt a family with only 7 animals is eating their own regularly enough that it is reflected in a 7-day recall. Give some additional thought to how to word this sentence so it better reflects what you say in the rest of the paragraph. starting line 164. While I agree that your data for milk suggests the value for cattle production (even when you have not corrected for milk producers), the data for eggs do not – you need to address this inverted U-shaped curve. line 197-198. I am not sure the inclusion of expenditures made these results independent of income generation effects. You do not appear to have income generation nor know where the money came from for the market foods. I would rephrase this. Your final statement does not seem to match the results. The association with production was not consistent – as it was with market expenditures. Also, you do not have a measure of nutritional status – you have a measure of diet. I would reconsider your conclusion. Ethics approval and method of consent need to be added to the Methods section.

Reviewer 3 Report

This article is quite interesting, although it does not bring very new information. rather, it summarizes already known information. contains numerous stylistic and factual errors. additional research was carried out 5-6 years ago, so it seems to me that before publication it should be slightly updated and only to draw conclusions. compilation of results from 2014 regarding the consumption of various products basically has no reference to the current situation and trends. it is so dynamically changing in nutrition that such publications should be the most up-to-date.
